# The Clinical and Medico-Legal Aspects in the Challenge of Transfusion-Free Organ Transplants: A Scoping Review

**DOI:** 10.3390/jcm14155444

**Published:** 2025-08-01

**Authors:** Matteo Bolcato, Ludovico Fava, Aryeh Shander, Christoph Zenger, Kevin M. Trentino, Mario Chisari, Vanessa Agostini, Ivo Beverina, Giandomenico Luigi Biancofiore, Vincenzo De Angelis

**Affiliations:** 1Department of Medicine, Saint Camillus International University of Health and Medical Sciences, 00131 Rome, Italy; matteo.bolcato@unicamillus.org; 2MESIT Foundation of Social Medicine and Innovation Tecnology, 00131 Rome, Italy; 3Department of Anesthesiology and Critical Care, Englewood Health, Englewood, NJ 07631, USA; aryeh.shander@ehmchealth.org; 4Center for Health Law and Management, University of Bern, 3012 Bern, Switzerland; christoph.zenger@rw.unibe.ch; 5Medical School, The University of Western Australia, Perth, WA 6009, Australia; kevin.trentino@uwa.edu.au; 6Rodolico—San Marco Hospital, 95123 Catania, Italy; m.chisari@policlinico.unict.it; 7Transfusion Medicine, IRCCS Ospedale Policlinico San Martino, 16132 Genoa, Italy; vanessa.agostini@hsanmartino.it; 8Transfusion Medicine, ASST Valle Olona, 21052 Busto Arsizio, Italy; ivo.beverina@asst-valleolona.it; 9Transplant Anesthesia and Critical Care, University Hospital of Pisa, 56121 Pisa, Italy; giandomenico.biancofiore@unipi.it; 10National Blood Centre, Italian National Institute of Health, 00131 Rome, Italy; vincenzo.deangelis@iss.it

**Keywords:** organ transplantation, patient blood management, bloodless, Jehovah witnesses, blood transfusion

## Abstract

**Background**: Patient blood management (PBM) strategies have been shown to significantly reduce the use of blood products and enabled surgical procedures to be carried out safely without the need for transfusions. This evidence has raised questions about the possibilities of the “extreme” application of PBM strategies for complex surgical interventions, such as organ transplants, even in patients in whom it is not possible to proceed with transfusion. The aim of this scoping review was to identify and describe the current evidence available in the medical literature on the transplant of the four main solid organs: kidney, heart, liver, and lung in patients declining blood transfusions. **Methods**: A comprehensive literature search was conducted using PubMed from January 2000 to February 2025. Only articles reporting cases, case series, population samples, or comparative studies describing solid organ transplantation without the use of blood components were included. The results are presented separately for each solid organ. **Results**: Kidney: Nine studies were included, seven of which reported case reports or case series of kidney or kidney–pancreas transplants, and two articles were comparative studies. Liver: Nine studies reported bloodless liver transplants, eight were case reports or case series, and one was a comparative observational study. Heart: Five studies were included, four of which were case reports of heart transplants; in addition there was a comparative study describing eight heart transplants without the use of blood components to 16 transfusable transplant patients. Lung: Five studies reporting lung transplant without transfusion were reported, four of which were case reports performed in the absence of deaths, and two of which were bilateral. Furthermore, there was an article describing two single lung transplants without the use of blood components compared to ten transfusable transplant patients. **Conclusions**: The analysis performed demonstrates the possibility, depending on the organ, of performing solid organ transplant procedures without the use of blood components in selected and carefully prepared patients by experienced multidisciplinary teams.

## 1. Introduction

In recent decades, there has been a growing interest in the topic of blood transfusion outcomes, the sustainability of the transfusion system and the opportunities for saving the use of blood components through the application of the patient blood management (PBM) programs [1,2]. PBM has established itself as the standard of care in transfusion medicine, applied to all medical and surgical specialties with the aim of increasing patient safety [3,4,5].

The implementation of PBM has been shown to significantly reduce the use of blood products and enables many surgical procedures to be carried out safely without the need for transfusions [6,7]. This evidence has raised questions about the possibilities of an “extreme” application of this program for the performance of complex interventions, such as organ transplants, in patients in whom it is not possible to proceed with transfusion, such as those who, for ethical and religious reasons, refuse blood transfusions [8,9].

Transplantation is a surgical procedure that consists of replacing a diseased and non-functioning organ or tissue with a healthy one of the same type, coming from another living or deceased donor. Transplantation is a life-saving treatment (heart, liver, lung) that substantially modifies the quality of life of a patient (kidney).

Performing organ transplantation without resorting to blood transfusions is particularly requested by the community of Jehovah’s Witnesses, who are numerically well represented in Europe and the United States. Hospitals and institutions have responded inconsistently to this request, often overlooking or unable to access data from the latest scientific literature. Even on a medico-legal and ethical level, discordant positions have been recorded by several hospitals in the same country, without a complete analysis of the problem and the evidence currently available.

The aim of this manuscript is to identify the current evidence and experiences present in the world literature on the transplantation activities of the four main solid organs: kidney, heart, liver, and lung.

## 2. Materials and Methods

This literature survey followed the guidelines of the Preferred Reporting Items for Systematic Reviews and Meta-Analyses (PRISMA) framework for the preparation and reporting for scoping reviews [10] (see Appendix A).

The articles identified in the present review were sourced from the PubMed databases. For the search strategy, we decided to use the following keywords (OR): Bloodless [kidney-heart-liver-lung] transplant; OR [kidney-heart-liver-lung] transplant without blood; OR Jehovah’s Witness [kidney-heart-liver-lung] transplant; OR refuse blood [kidney-heart-liver-lung] transplant; OR decline blood [kidney-heart-liver-lung] transplant; OR transfusion free [kidney-heart-liver-lung] transplant.

Eligibility criteria: The results present in the following time interval 1 January 2000 to 28 February 2025 were searched. We restricted results to the English language and to full-text and abstract publications.

The research reviewers, in three couples, carried out the initial search for the papers. They used the search protocol described above to identify the relevant publications. In the case of disagreements, the consensus of the research supervisors was definitive. The researchers used the following research order: titles were screened first and then the abstracts and full papers. A paper was considered potentially relevant and its full text reviewed if, following discussion between the two independent reviewers, it could not be unequivocally excluded on the basis of its title and abstract. The full text of all papers not excluded based on the abstract or title was evaluated.

Only articles reporting cases, case series, population samples, or comparative studies describing solid organ transplantation without the use of blood components were included.

## 3. Results

The results are presented separately for each solid organ.

### 3.1. Kidney

After our search process for kidney, we retrieved a total of 40,407 results. After the in-depth screening, nine studies were included, seven of which reported case reports or case series of kidney or kidney–pancreas transplants (Table 1). In addition to these, there are two articles that compare the group of non-transfusable subjects with those who were transfused (Table 2).

All studies indicated successful transplants in the absence of deaths or subsequent transfusion needs. The findings in the literature showed substantial stability of the data with experiences spread in different parts of the world with the same chances of success regardless of whether the kidney transplant also involved the pancreas or not. Some studies underline the importance of adequate patient preparation and the use of erythropoietin and other tools specific to PBM. For this type of transplant, there are quite a number of comparative experiences.

**Table 1 jcm-14-05444-t001:** Non-comparative studies in kidney transplants.

Author	Year	Country	Study Type	Population	Transplant	Donor Type	Main Findings
Figureiro J. et al. [11]	2003	USA	Case series	5 adults	KPT	DD	Two patients experienced acute rejection at some point after transplantation. Three developed chronic rejection eventually resulting in ESRD. Patient survival was 100%. Graft survival was 60% for the kidney and 100% for the pancreas, with a range of 6–96 months and mean of 4 yr.
Boggi U. et al. [12]	2004	Italy	Case series	6 adults	3 KTA, 1 KPT, 1 DKT, 1 PTA	DD	After a mean follow-up of 31.4 months (range: 18 to 39) all the recipients are alive and well with functioning grafts. One recipient required blood transfusions 1 month after grafting due to drug-related toxicity.
Hernandez Navarrete LS. et al. [13]	2013	Mexico	Case series	3 adults;	3 KTA	DD	One of the patients had a perirenal hematoma as a complication, which required surgery 20-day post-transplant. At 5-, 26-, and 36-months post-transplant, the three patients were alive and with functional grafts.
Spasovski G. et al. [14]	2014	Macedonia	Case study	1 adult	1 KTA	LD	While performing ureter–bladder anastomosis, sudden bleeding occurred from a kidney rupture. The patient was immediately infused with 1.5 L of crystalloids, an appropriate surgical intervention; the patient was not in an extreme need for blood transfusions, having an appropriate cardiovascular compensation.
Gomez MF. et al. [15]	2017	USA	Case study	1 adult	1 KPT	DD	First case of the use of a hemoglobin-based oxygen carrier (HBOC) transfusion in a double solid organ transplant patient.
Miyake K. et al. [16]	2019	Japan	Case series	3 adults	3 KTA	LD	No complications observed.
Guerra G. et al. [17]	2020	USA	Case study	1 adult	1 KTA	DD	Post-operative course complicated by severe anemia (day 4). Patient and family refused transfusion.

KPT: kidney–pancreas transplant; KTA: kidney transplant alone; DKT: dual kidney transplant; PTA: pancreas transplant alone; DD: deceased donor; LD: living donor.

**Table 2 jcm-14-05444-t002:** Comparative studies in kidney transplants.

Author	Year	Country	Study Type	Population	Transplant	Main Findings
Cumminis PJ. et al. [18]	2018	USA, Italy	Case report, Review of comparative study	Various	KTA (DD)	Outcomes for JW undergoing transfusion-free transplant were comparable to those of transplant patients who receive transfusions.
Carvahlo Fiel D. et al. [19]	2021	Portugal	Comparative study	143 JW patients (10 pediatric)/142 non-JW patients	KTA (DD)	No differences in the incidence of clinical indication for transfusion (13.3% versus 11.3%, *p* = 0.640), but a higher proportion of non-JW patients received transfusions (2.1% versus 9.2%, *p* = 0.010). No differences in the proportion of patients with decreased hemoglobin concentration, in reinterventions due to hemorrhagic complications, in the use of erythropoiesis-stimulating agents at hospital discharge, in the incidence of acute rejection, in renal function, and in the mortality or graft survival rate at 12 months.

JW: Jehovah’s Witnesses; KTA: kidney transplant alone; DD: deceased donor.

### 3.2. Liver

After our search process for liver, we retrieved a total of 28,632 results. After the in-depth screening, we identified nine studies reporting bloodless liver transplants. Of these, eight were case reports or case series (Table 3) and one was a comparative observational study (Table 4) of non-transfusable and transfusable patients.

The eight case reports and series published between 2000 and 2020 consisted of fifty-four bloodless patients, three of which were pediatric. These studies originated from the US, Australia, Canada, Korea, Belgium, and Italy. Hospital mortality was the only outcome reported by all case reports and series. The other outcomes reported included complications, long-term mortality, hospital length of stay (LOS), and intensive care unit (ICU) LOS.

Overall, of the fifty-four patients included in the case reports and series, four (7.4%) died in hospital. One of the in-hospital deaths was in a patient whose family requested a transfusion after the patient’s postoperative hematocrit was 8.2% (hemoglobin level 2.73 g/dL). Long-term mortality, ranging from 1 years to 5 years, was reported by seven of the case reports and series, with an event rate of 5 of 53 (9.3%). Complications (including peritonisis, arrythmia, acute kidney injury [AKI], reperfusion syndrome, urinary tract infection [UTI], and reoperation) were included as an outcome in five studies and the event rate was 7 of 26 (26.9%). Three case series reported a mean hospital LOS of 20.0 days, while two case series reported a mean ICU LOS of 4.2 days.

When analyzed separately, the three pediatric patients included in the case reports and series involved liver transplants in a 6-month-old and 3-year-old in the US and a 6-year-old in Belgium. None of the patients died in hospital or at 33 months in one study (Jabbour 2005) [20] or 41 months in another study (Detry 2005) [21]. In the US study, the two pediatric patients did not experience any intraoperative complications and had a hospital LOS of 14 and 18 days. The 6-year-old patient from Belgium developed peritonitis secondary to perforated gastric ulcer on day 6 and was reoperated. The patient eventually received one unit of red cells (against the parents’ will) at a hematocrit of 16% (hemoglobin level 5.3 g/dL) and had a LOS of 32 days.

The comparative observational study originated in the US and reported the results from comparing thirty transfusion-eligible to eight transfusion-free adult patients. No complications were reported in either group, while reoperations occurred in two (25%) patients in the transfusion-free group and nine (30%) patients in the transfusion-eligible group. Patients in the transfusion-free group had a mean ICU LOS of 5 days (range: 2–9) and a mean hospital LOS of 17.5 days (range: 8–27) compared to 5.8 (range: 2–18) and 19.5 (range: 8–71) in the transfusion-eligible group. Long term mortality was 0% in the transfusion-free group and 10% in the transfusion-eligible group, with mean follow-up times of 817 and 672 days respectively.

**Table 3 jcm-14-05444-t003:** Non comparative studies in liver transplant.

Author	Year	Country	Study Type	Population	Intervention	Outcomes	Results
Baldry C. et al. [22]	2000	Canada	Case study	1 adult	DDLT (not clear)	Complications	0%
						Hospital mortality	0%
Jabbour N. et al. [23]	2004	US	Case study	1 adult	two-stage LDLT	Complications	0%
						Hospital mortality	0%
						Long term mortality	0%
Detry O. et al. [21]	2005	Belgium	Case series	8 adults; 1 pediatric	DDLT: 6 adults and 1 pediatric; LDLT: 2 adults	Severe complications	22%
						Hospital mortality	10%
						Long term mortality	10%
Jabbour N. et al. [20]	2005	US	Case series	24 adults	LDLT: 66.7% DDLT: 33.3%	Hospital mortality	8%
						30-day mortality	8%
						Long term mortality	8%
						Hospital LOS, mean	20.5
Jabbour N. et al. [24]	2005	US	Case series	2 pediatric	LDLT	Complications	0%
						ICU LOS, mean	4.0
						Hospital LOS, mean	16.0
						Hospital mortality	0%
						Long term mortality	0%
Jeffrey G. et al. [25]	2007	Australia	Case series	2 adults	DDLT	Hospital mortality	0%
						Long term mortality	0%
Jeong J. et al. [26]	2017	Korea	case series	2 adults	LDLT	Hospital mortality	50%
						Long term mortality	50%
Costanzo D. et al. [27].	2020	Italy	Case series	13 adults	DDLT	ICU LOS, mean	4.2
						Hospital LOS, mean	19.7
						AKI	31%
						Reperfusion syndrome	15%
						Re-operation	15%
						UTI	8%
						Hospital mortality	0%
						30-day mortality	0%
						1-year mortality	8%

DDLT: deceased donor liver transplant; LDLT: living donor liver transplant; Hospital and ICU LOS (length of stay) are reported in days.

**Table 4 jcm-14-05444-t004:** Comparative study in liver transplant.

Author	Year	Country	Study Type	Population	Intervention	Outcomes	Transfusion-Free	Transfusion-Eligible
Jabbour N. et al. [28]	2004	US	Comparative study	38 adults (8 in the transfusion-free group; 30 in the transfusion-eligible group)	LDLT	Complications	0%	0%
						ICU LOS (SD)	5.0 (2.3)	5.8 (4.1)
						Hospital LOS (SD)	17.5 (7.3)	19.5 (13.3)
						Reoperation	25%	30%
						Long term mortality	0%	10%

LDLT: living donor liver transplant; Hospital and ICU LOS (length of stay) are reported in days; SD: standard deviation.

### 3.3. Heart

After our search process for heart transplants, we retrieved a total of 19,346 results. After the in-depth screening, five studies were included, four of which were case reports of heart transplants that reported no deaths (Table 5). Furthermore, one article described eight heart transplants without the use of blood components through a comparison of the study (case-control) between non-transfusable and transfusable patients. This study did not detect negative outcomes in the non-transfused patients (Table 6).

### 3.4. Lungs

After our search process for lung, we retrieved a total of 19,924 results. After the in-depth screening, five studies were included, four of which represented case reports of lung transplants performed in the absence of deaths (Table 7), two of which were bilateral; furthermore, there was an article that described two single lung transplants without the use of blood components through a comparison between the group of non-transfusable subjects and those transfused without detecting particular negative outcomes in the non-transfused (Table 8).

## 4. Discussion

### 4.1. Kidney

Renal transplantation consists of a surgical operation during which a kidney obtained from a donor (deceased or living) is implanted inside the body of the recipient. In this case, kidney transplantation is a heterotopic transplant, meaning the organ is positioned in a different location than the native organs, and therefore, except in particular cases or specific conditions, the native organs remain in situ. Intraoperative mortality is close to zero. The new kidney is usually positioned in the right iliac fossa, the arteries and veins of the new kidney are joined to the vessels of the recipient and the ureter is connected to the bladder. The characteristics of the intervention are objectively much less destructive than other types of transplants and, as a rule, do not require removal of the recipient’s kidney. As a result, there is less risk of bleeding and the transfusion rate in this type of transplant is around 45% [39]. From this data, performing a kidney transplant without the use of allogeneic blood components is realistically achievable. In addition, avoiding the administration of blood products improves the outcomes of transplanted patients. In conclusion, achieving kidney transplants without the use of transfusions can be achieved with the widespread and careful application of PBM principles [40].

These data suggest that any ethical objections to bloodless kidney transplant can be overcome and perpetuating dialysis in patients who are potentially eligible can be avoided. These actions will likely result in significantly improvement in patient care and quality of life.

### 4.2. Liver

Liver transplantation is the only therapeutic option for patients with end-stage liver disease or acute liver failure due to viral infections, alcohol abuse, cancer, congenital diseases, and intoxications. The procedure has seen numerous and significant improvements in recent years. In particular, due to several multimodal and multidisciplinary efforts aimed at reducing perioperative bleeding, liver transplants without the use of blood products is now a reality. In fact, during the last decade, blood product requirements in liver transplant patients have significantly decreased in most centers and this improvement was related to different factors including better surgical techniques, liver transplant indication, and liver graft preservation techniques. Also, the experience of the surgical and anesthesiologic team is important.

Liver transplantation is a demanding procedure from the surgical and anesthesiology perspectives with an average post-transplant one-year survival of around 87% and a perioperative mortality between 5% and 10%.

However, available studies indicate that a blood-free liver transplant is feasible and that its risk-to benefits ratio can be maintained in selected adult patients provided it is carried out at experienced centers and by applying a multidisciplinary approach.

The findings in the literature show a substantial stability of the data with widespread experiences in various parts of the Western world from the early 2000s onwards.

Our review identified one comparative study of bloodless liver transplants to a transfusable population. The results of this study suggested bloodless liver transplant can be performed safely with comparable outcomes.

### 4.3. Heart

A heart transplant is a complex surgical procedure aimed at replacing a diseased heart with a healthy one taken from a donor. The clinical indication is terminal heart disease for which there is no medical therapy or other surgical intervention. A life expectancy of less than 6–12 months occurs in a patient with a chronological and physiological age usually less than 60 years who is not burdened by other systemic diseases. The pathological indications consider cardiac diseases that irreversibly compromise ventricular muscle function and therefore contractility, and includes primary cardiomyopathies, secondary cardiomyopathies, and complex congenital cardiomyopathies.

It should be considered that this type of transplant is particularly complex with a non-negligible mortality; according to data in the literature, on average, subjects who were candidates for transplants showed a one-year survival rate of around 81% with a perioperative mortality rate just under 10% [41].

In the case of heart transplants without the use of blood components, the limited number of available publications reduce the possibility of making a recommendation based on proof of evidence. It is, however, possible to consider that from the currently available evidence, heart transplants without the use of blood transfusions are possible in carefully selected patients and subjected to an optimization of the conditions by a multidisciplinary team before the procedure. With these assumptions, from the data available to date, there are no significant differences in mortality, loss of the transplanted organ, or rejection compared to the transfused population.

### 4.4. Lung

Lung transplantation is a surgical procedure that involves replacing a diseased non-functioning lung with a healthy one. The procedure may involve only one or both lungs. Indications for lung transplantation may include all non-neoplastic terminal lung diseases as long as the patient does not have other serious pathologies that would prevent chronic immunosuppressive therapy. Although bilateral lung transplants and heart transplants are possible, in most cases, a single lung transplant is performed that results in sufficient improvement in lung function for each of the two recipients of the organs from a single donor. The main indications for lung transplantation include chronic obstructive pulmonary disease (COPD), idiopathic pulmonary fibrosis, cystic fibrosis, idiopathic pulmonary hypertension, and alpha-1 antitrypsin deficiency.

The surgical procedure is the most complex and burdened by a significant risk of mortality among solid organ transplants; perioperative mortality is around 15% and survival one year after transplant is not higher than 80% [42].

Regarding the possibility of performing lung transplants without the use of blood components, the small number of publications and experiences present in the literature do not allow for a detailed analysis at this point. It must be considered that although this type of transplant is particularly complex and burdened by a significant mortality rate, it has been performed several times in various countries without using the transfusion of allogeneic blood. In the only comparative study present, no application criticalities were detected in the carefully selected patients who underwent suitable preparation before performing the procedure.

### 4.5. General

This review shows the current evidence found in the literature regarding the possibility of performing solid organ transplants without the use of blood components. From these data, kidney transplantation appears to be the safest and easiest to be performed without transfusion and has the most published articles. It is likely the application of PBM strategies along with improvements in surgical techniques and devices will contribute to a gradual and sustained reduction in transfusion use across the entire kidney transplant population, including in patients accepting transfusions.

By contrast, lung and heart transplantation are characterized by a paucity of literature, probably also due to the smaller number of events themselves and their complexity. These interventions represent the smallest number of publications in our review, suggesting many centers would decline performing them without transfusions. With regards to liver transplantation, the procedure is complex but the literature strongly suggests the possibility of performing such a transplant without transfusions.

Significant considerations can also be made on an ethical and medico-legal level following the analysis of the current evidence. A priori denial of transplanting a solid organ or the exclusion of a patient who declines transfusion on the transplant list does not appear in harmony with the fundamental bioethical principle of justice. In fact, the technical impossibility of carrying it out or the abstract excessiveness of the risk incurred cannot be supported a priori; on the contrary, it appears legally correct and ethically necessary to carefully evaluate the individual patient and weigh the potential and indications coming from the application of PBM strategies. In complex cases, it is also possible to direct the patient to a center better able to perform these procedures with the least possible risk.

On this point in the context of European jurisprudence, what is indicated by the Judgment of the Court of 29 October 2020, A v. Veselības ministrija, C-243/19, ECLI:EU:C:2020:872 is of interest, in which the Court of Justice of the European Union has—in summary—indicated that a citizen of the Union has the possibility of receiving an authorization from his country of origin to receive treatment abroad, in another member state, if those offered are discriminatory in terms of the religious conscience of the subject.

### 4.6. Strengths and Weaknesses of the Study

To our knowledge, this is the first time a comprehensive review has been conducted covering the four transplantable solid organs. We believe this comprehensive data, compiled over a 25-year period, is a strength and can be useful for gaining an understanding of the current literature regarding the feasibility of performing solid organ transplants without the use of blood component support. However, there are numerous limitations to our study.

The number of available studies is limited, and, particularly for heart and lung transplants, comparative studies are limited in number and are not identically constructed or have the same study design. For example, some did not indicate which PBM strategies (if any) were used in the control group. Regarding the observational studies included in this study, in some of them, we did not have information regarding any procedures attempted on patients that resulted in a negative outcome, which may constitute a bias in this type of study.

Ultimately, there are objective limitations of the data collection in this study, but the importance of the issue and scientific interest demonstrate the need for further systematic studies, especially in centers that continuously and systematically apply PBM programs to all patients.

## 5. Conclusions

Our study demonstrates the possibility, depending on the organ, of performing solid organ transplant procedures, without the use of blood components, in selected and carefully prepared patients by experienced multidisciplinary teams. It seems foreseeable that, with the expansion of PBM implementation globally, safe techniques and programs will be increasingly available to successfully perform these surgeries in those who decline transfusions as well as those who accept them.

## Figures and Tables

**Table 5 jcm-14-05444-t005:** Non-comparative studies in heart transplants.

Author	Year	Country	Study Type	Population	Main Findings
Sue SH et al. [29]	2008	China	Case report	1 adult	Orthotopic heart transplantation successfully performed in a 46-year-old man with congenitally corrected transposition of the great arteries. No experienced rejection and is doing well in post-transplant year 4.
Russo MJ et al. [30]	2013	USA	Case report	1 adult	First case of JW patient undergoing a second orthotopic heart transplantation 20 years after his first transplant. On postoperative day 1, the patient experienced a cardiac arrest with pulseless electrical activity (PEA). A chest x-ray suggested the presence of right pleural bleeding, leading to emergent chest exploration. Extracorporeal membrane oxygenation (ECMO) was initiated. While on ECMO support, the patient’s course was further complicated by anuric renal failure, requiring continuous venovenous hemodialysis and a decreased hemoglobin level of 4.7 mg/dL. The patient was discharged to home on POD 52/HD 60 with a feeding tube. At discharge, the hemoglobin was 10.3 mg/dL. The tracheostomy was removed, and the renal function recovered. At further follow-up (non-specified), there was no evidence of rejection.
Dallas T et al. [31]	2015	USA	Case report	1 adult	With a multidisciplinary team, preoperative erythropoietin-stimulating drugs, normovolemic hemodilution, cell salvage, and pharmacotherapy to prevent and treat coagulopathy, it was possible to maintain hemoglobin levels greater than 11 g/dL (lowest level 11.3 g/dL). Patient was discharged on post-operatory day 10 in stable conditions. At 2 months follow-up, no symptoms of HF and FE 40% was present.
Tsukioka Y et al. [32]	2024	USA	Case report	1 adult	Heart transplantation in a 68-year-old Jehovah’s Witnesses patient with congenitally corrected transposition of the great arteries who developed heart failure due to right ventricular dysfunction. The postoperative course was uneventful. Hemoglobin levels consistently exceeded 11 g/dL. Postoperative right heart catheterization study demonstrated significant improvement in pulmonary artery pressure, pulmonary capillary wedge pressure, and cardiac output.

**Table 6 jcm-14-05444-t006:** Comparative study in heart transplant.

Authors	Year	Country	Study Type	Population	Main Findings
Sander S. et al. [33]	2021	USA	Comparative study	8 JW patients/ 16 controls	A very careful selection of JW patients and pre-operative optimization with a correction of iron deficiency and erythropoietin was conducted. JW recipients had higher hemoglobin levels at transplant (13.5 mg/dL vs. 12.6 mg/dL). There was no observed difference in short- or long-term outcomes, including in primary graft dysfunction, length of stay, treated rejection, cardiac allograft vasculopathy, or death. Study limited by the retrospective data analysis and the small sample.

JW: Jehovah’s Witnesses.

**Table 7 jcm-14-05444-t007:** Non-comparative studies in lung transplants.

Authors	Year	Country	Study Type	Population	Condition	Intervention	Main Findings
Grande A.M. et al. [34]	2003	Italy	Case report	1 adult	IPF	SLT	Singe left LT was performed in 38 y.o. JW woman affected by IFP. Total ischemic time was 255 min. The patient was extubated, and norepinephrine infusion was stopped on 1st POD. The subsequent course was regular, and the patient was transferred to the Pneumology Dept. on 12th POD. Patient died 14 months after LT for mild acute rejection with multiple thromboembolisms and pulmonary infarction.
Cerezo Madueno F. et al. [35]	2013	Spain	Case report	1 adult	COPD	SLT	A right single LT was performed, with an ischemic time of 340 min. No extracorporeal circulation was required, and there were no intraoperative complications. The patient remained in the hospital for 35 days. He presented the following complications: an episode of acute rejection and pneumonia. At 18 months of follow-up, the patient was asymptomatic, with a good quality of life and performance status. The patient has since died, 25 months after receiving the transplant, having developed obstructive bronchial syndrome 7 months before death.
Fernandez Tujillo L. et al. [36]	2020	Colombia	Case report	1 adult	IPF	SLT	Transplantation was performed without complications nor blood product requirement, intraoperative cell salvage was performed, and pharmacological agents were used preoperatively for bleeding prevention. The patient only developed anemia after administration of immunosuppressor therapy, which was treated with erythropoietin in the outpatient setting.
Chan EG. et al. [37]	2021	USA	Case series	2 adults	1 IIF, 1 COPD	BSLT	First patient experienced respiratory inefficiency requiring intubation on POD 1 for pulmonary edema. Patient still alive with no respiratory limitation at 107 months follow-up. Second patient had a post-op course complicated with aspiration pneumonia and unfortunately suffered a severe myocardial infarction after a surveillance procedure at 63 months and passed away. No evidence of chronic rejection.

IPF: interstitial lung disease; COPD: chronic obstructive lung disease; IIF: inhalation injury fibrosis; SLT: single lung transplant; BSLT: bilateral sequential lung transplant; POD: post-operative day.

**Table 8 jcm-14-05444-t008:** Comparative studies in lung transplants.

Authors	Year	Country	Study Type	Population	Condition	Intervention	Main Findings
Partovi S. et al. [38]	2013	Switzerland	Comparative study	2 JW/10 controls	IPF	SLT	Postoperative FEV1 and FVC were significantly higher in the JW group compared with the controls group (P 0.037 and P 0.036, respectively), but probably an incidental observation related to small sample size. No significant difference for the length of stay in ICU or in the hospital (P 0.437 and P 0107).

JW: Jehovah’s Witnesses; IPF: interstitial lung disease; SLT: single lung transplant.

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
