# Peer review of "The Clinical and Medico-Legal Aspects in the Challenge of Transfusion-Free Organ Transplants: A Scoping Review"

_jcm, 2025, doi:10.3390/jcm14155444_

Round 1
Reviewer 1 Report
Comments and Suggestions for Authors
Thank you for allowing me to review this interesting manuscript. The authors aim to provide an overview of bloodless transplantation surgery outcomes for kidney, liver, heart, and lung by reviewing the relevant published articles. They conclude that kidney transplants are the easiest solid organ to transplant without blood transfusion. While liver, heart, and lung transplants are increasingly complex, the authors indicate that certain patients can be managed bloodlessly. Overall, I applaud the authors’ efforts to advance the field of Patient Blood Management, and I hope the authors and editors find my comments helpful.
- Not the focus of my review, but grammatical and syntax issues throughout the manuscript could be edited. Additionally, Lines 165 and 175 state “liver” rather than “heart” and lung”, respectively.
- Liver, section 3.2, tables 3 and 4. Regarding bloodless liver transplants, were any differences in outcomes noted between deceased-donor (DDLT) versus living-donor (LDLT) transplantations?
- Liver, section 3.2, tables 3 and 4. It is known that one benefit of living-donor transplantation is taking the patient to transplant at a lower MELD score than for DDLT. A comparison of MELD scores and outcomes between LDLT and DDLT could be informative.
- Discussion, section 4.5. PBM incorporates pre-operative, intra-operative, and post-operative strategies to minimize blood loss and optimize oxygen carrying capacity. A detailed discussion of the pre-op, intra-op, and post-op strategies used in the reviewed reports would be valuable. Was pre-op anemia corrected? Were antifibrinolytics employed during surgery? Was autologous blood used (pre-op autologous donation, acute normovolemic hemodilution, RBC salvage/washing)?
Light editing for grammar and syntax would improve the paper's readability.
Author Response
Thank you for appreciating our work.
1. Not the focus of my review, but grammatical and syntax issues throughout the manuscript could be edited. Additionally, Lines 165 and 175 state “liver” rather than “heart” and lung”, respectively.
We thank the Reviewer for their comments. We have corrected the error on lines 165 and 175. In addition to this we have noted the Reviewer’s comment on grammatical and syntax issues and had a native English-speaking co-author provide further edits. We believe these edits have resulted in a significantly improved submission.
2. Liver, section 3.2, tables 3 and 4. Regarding bloodless liver transplants, were any differences in outcomes noted between deceased-donor (DDLT) versus living-donor (LDLT) transplantations?
The articles identified in the literature on bloodless liver transplants did not compare outcomes between DDLT and LDLT. As a result we are unable to comment on differences in outcomes.
3. Liver, section 3.2, tables 3 and 4. It is known that one benefit of living-donor transplantation is taking the patient to transplant at a lower MELD score than for DDLT. A comparison of MELD scores and outcomes between LDLT and DDLT
could be informative.
Thank you. We agree with this statement, but some studies do not report this data, and we prefer not to delve into just one organ in this article. However, in future articles, we would like to publish more in-depth articles on individual organs.
4. Discussion, section 4.5. PBM incorporates pre-operative, intra-operative, and post- operative strategies to minimize blood loss and optimize oxygen carrying capacity. A detailed discussion of the pre-op, intra-op, and post-op strategies used in the
reviewed reports would be valuable. Was pre-op anemia corrected? Were antifibrinolytics employed during surgery? Was autologous blood used (pre-op autologous donation, acute normovolemic hemodilution, RBC salvage/washing)?
We would like the thank the Reviewer for this good suggestion. Unfortunately, we found the reporting of PBM strategies highly variable, with some articles not reporting much detail. As a result, we did not include this information in the Table. For this reason we have made a general consideration in the considerations section. We hope you can understand. Many thanks!
Reviewer 2 Report
Comments and Suggestions for Authors
Review Report
Manuscript Title: The challenge of transfusion free organ transplant. Clinical and Medico Legal aspects: A scoping review
Manuscript Type: Review Article
Brief Summary
This manuscript presents a scoping review of the literature from 2000 to 2025 on the feasibility of performing solid organ transplantation (kidney, liver, heart, lung) without the use of allogeneic blood transfusion. It focuses on patients who decline transfusion, especially Jehovah’s Witnesses, and analyzes clinical outcomes along with the ethical and medico-legal implications of performing or denying transplantation in these patients.
General Concept Comments
- Relevance and structure
The topic is timely and of high clinical, ethical, and legal relevance. The manuscript is clearly structured by organ type, and the inclusion of medico-legal analysis adds interdisciplinary value.
- Major methodological limitations not sufficiently addressed
- Inadequate comparative framework
The included comparative studies do not focus specifically on patients with a documented clinical indication for transfusion (e.g., severe perioperative anemia). Instead, they compare cohorts based on transfusion acceptance or refusal, regardless of actual transfusion need. This limits the interpretability of the outcome data.
The authors should explicitly acknowledge this limitation and emphasize the need for future studies focused on high-risk patients who develop transfusion-requiring anemia. - Asymmetric application of Patient Blood Management (PBM)
PBM strategies (e.g., erythropoietin, iron therapy, normovolemic hemodilution, cell salvage) appear to have been used preferentially or exclusively in patients who refused transfusion. It is unclear whether these interventions were applied in the transfusion-eligible control groups. This introduces a bias that may explain comparable outcomes.
The manuscript should address this imbalance and discuss its implications when interpreting the data. - Lack of outcome data in patients with life-threatening anemia
The review does not provide detailed information on cases where patients with critical anemia declined transfusion. Except for a single pediatric case in Belgium, there is insufficient analysis of outcomes in patients who died or experienced major complications that might have been preventable with transfusion.
The authors should highlight this data gap and encourage systematic documentation of such cases in future literature.
- Ethical and medico-legal considerations
While the manuscript discusses the principle of justice and the rights of transfusion-refusing patients, it does not address a key ethical issue: whether refusal of transfusion leads to preventable harm in patients with clear clinical indications for it (e.g., severe anemia with bleeding). A meaningful ethical analysis should compare outcomes in transfusion-refusing vs. transfusion-accepting patients under comparable risk conditions.
A dedicated paragraph or sub-section should address this ethical tension between autonomy and non-maleficence. This could be framed using published data or formal ethical frameworks. Without this, the ethical dimension of the review remains incomplete.
Additionally, the legal section could benefit from a brief comparative mention of how other healthcare systems (e.g., US, Germany, France) approach similar dilemmas.
- Writing, references, and figures
The manuscript is well written, with a clear and concise style. The references are appropriate, relevant, and mostly recent. The tables and figures are adequate and informative. There is no overuse of self-citation.
Specific Comments
- Lines 24–27: Clarify that this is a scoping review, not a systematic review or meta-analysis.
- Tables 4 and 6: Indicate whether PBM strategies were also implemented in the control groups for accurate comparison.
- Section 4.5 (Lines 271–284): Consider subdividing this section for clarity and include a call for prospective, multicenter studies with stratified risk cohorts.
- Conclusions (Lines 302–308): Recommend distinguishing between feasibility in selected individuals and broader clinical applicability.
Author Response
Brief Summary
This manuscript presents a scoping review of the literature from 2000 to 2025 on the feasibility of performing solid organ transplantation (kidney, liver, heart, lung) without the use of allogeneic blood transfusion. It focuses on patients who decline transfusion, especially Jehovah’s Witnesses, and analyzes clinical outcomes along with the ethical and medico-legal implications of performing or denying transplantation in these patients.
General Concept Comments
1. Relevance and structure
The topic is timely and of high clinical, ethical, and legal relevance. The manuscript is clearly structured by organ type, and the inclusion of medico-legal analysis adds interdisciplinary value.
1. Major methodological limitations not sufficiently addressed
1. Inadequate comparative framework
The included comparative studies do not focus specifically on patients with a documented clinical indication for transfusion (e.g., severe perioperative anemia). Instead, they compare cohorts based on transfusion acceptance or refusal, regardless of actual transfusion need. This limits the interpretability of the outcome data. The authors should explicitly acknowledge this limitation and emphasize the need for future studies focused on high-risk patients who develop transfusion-requiring
anemia.
2. Asymmetric application of Patient Blood Management (PBM) PBM strategies (e.g., erythropoietin, iron therapy, normovolemic hemodilution, cell salvage) appear to have been used preferentially or exclusively in patients who refused transfusion. It is unclear whether these interventions were applied in the transfusion-eligible control groups. This introduces a bias that may explain comparable outcomes. The manuscript should address this imbalance and discuss its implications when interpreting the data.
3. Lack of outcome data in patients with life-threatening anemia. The review does not provide detailed information on cases where patients with critical anemia declined transfusion. Except for a single pediatric case in Belgium, there is insufficient analysis of outcomes in patients who died or experienced major complications that might have been preventable with transfusion. The authors should highlight this data gap and encourage systematic documentation of such cases in future literature.
We want to thank you for the excellent and accurate advice you gave us. We have tried to include this information in a new section specifically dedicated to the study's limitations. We believe this meets your expectations.
1. Ethical and medico-legal considerations. While the manuscript discusses the principle of justice and the rights of transfusion- refusing patients, it does not address a key ethical issue: whether refusal of transfusion leads to preventable harm in patients with clear clinical indications for it (e.g., severe anemia with bleeding). A meaningful ethical analysis should compare outcomes in transfusion-refusing vs. transfusion-accepting patients under comparable risk conditions. A dedicated paragraph or sub-section should address this ethical tension between autonomy and non-maleficence. This could be framed using published data or formal ethical frameworks. Without this, the ethical dimension of the review remains incomplete.
Additionally, the legal section could benefit from a brief comparative mention of how other healthcare systems (e.g., US, Germany, France) approach similar dilemmas.
We are aware that the issue raised is very large and important. We do not wish to conduct an ethical analysis of the problem, but simply to highlight the need to evaluate patients who refuse a transfusion on a case-by-case basis, depending on the organ to be transplanted, the experience of the transplant center, etc. We do not consider ourselves equipped to perform a complete ethical analysis of the issue, which, moreover, remains unresolved in many countries. We thank you for your suggestions, which we may provide for a future article.
1. Writing, references, and figures
The manuscript is well written, with a clear and concise style. The references are appropriate, relevant, and mostly recent. The tables and figures are adequate and informative. There is no overuse of self-citation.Specific Comments:
Thanks for your comments, we have provided the necessary changes as below.
Lines 24–27: Clarify that this is a scoping review, not a systematic review or meta-analysis.
We have made this update in the following locations Line 26-27
Tables 4 and 6: Indicate whether PBM strategies were also implemented in the control groups for accurate comparison.
Some articles do not contain specifications so it is impossible to give a detailed indication (see also the answer above).
Section 4.5 (Lines 271–284): Consider subdividing this section for clarity and include a call for prospective, multicenter studies with stratified risk cohorts.
We added these recommendations in the new paragraph: Strengths and weaknesses of the study.
Conclusions (Lines 302–308): Recommend distinguishing between feasibility in selected individuals and broader clinical applicability.
We indicated “(…) depending on the organ, (…) in selected and carefully prepared patients by experienced multidisciplinary teams”
Round 2
Reviewer 1 Report
Comments and Suggestions for Authors
Thank you for this revision and for addressing my previous comments and questions. I have no additional comments or questions to offer this revision.